# Molecular Hydrogen in Natural Mayenite

**Evgeny Galuskin [1],\*** , **Irina Galuskina [1]** , **Yevgeny Vapnik [2] and Mikhail Murashko [3]**

[1] Institute of Earth Sciences, Faculty of Natural Sciences, University of Silesia, Będzińska Street 60, 41-200 Sosnowiec, Poland; irina.galuskina@us.edu.pl

[2] Department of Geological and Environmental Sciences, Ben-Gurion University of the Negev, POB 653, Beer-Sheva 84105, Israel; vapnik@bgu.ac.il

[3] Faculty of Geology, Saint Petersburg State University, 7-9 Universitetskaya nab., 199034 St. Petersburg, Russia; mzmurashko@gmail.com

\* Correspondence: evgeny.galuskin@us.edu.pl

**Abstract:** In the last 15 years, zeolite-like mayenite, $Ca_{12}Al_{14}O_{33}$, has attracted significant attention in material science for its variety of potential applications and for its simple composition. Hydrogen plays a key role in processes of electride material synthesis from pristine mayenite: $\{Ca_{12}Al_{14}O_{32}\}^{2+}(O^2) \rightarrow \{Ca_{12}Al_{14}O_{32}\}^{2+}(e^-)_2$. A presence of molecular hydrogen in synthetic mayenite was not confirmed by the direct methods. Spectroscopy investigations of mayenite group mineral fluorkyuygenite, with empirical formula $(Ca_{12.09}Na_{0.03})_{\Sigma 12.12}(Al_{13.67}Si_{0.12}Fe^{3+}_{0.07}Ti^{4+}_{0.01})_{\Sigma 12.87}O_{31.96}$ $[F_{2.02}Cl_{0.02}(H_2O)_{3.22}(H_2S)_{0.15}\square_{0.59}]_{\Sigma 6.00}$, show the presence of an unusual band at 4038 cm$^{-1}$, registered for the first time and related to molecular hydrogen, apart from usual bands responding to vibrations of mayenite framework. The band at 4038 cm$^{-1}$ corresponding to stretching vibrations of $H_2$ is at lower frequencies in comparison with positions of analogous bands of gaseous $H_2$ (4156 cm$^{-1}$) and $H_2$ adsorbed at active cation sites of zeolites (4050–4100 cm$^{-1}$). This points out relatively strong linking of molecular hydrogen with the fluorkyuygenite framework. An appearance of $H_2$ in the fluorkyuyginite with ideal formula $Ca_{12}Al_{14}O_{32}[(H_2O)_4F_2]$, which formed after fluormayenite, $Ca_{12}Al_{14}O_{32}[\square_4F_2]$, is connected with its genesis. Fluorkyuygenite was detected in gehlenite fragments within brecciaed pyrometamorphic rock (Hatrurim Basin, Negev Desert, Israel), which contains reduced mineral assemblage of the Fe-P-C system (native iron, schreibersite, barringerite, murashkoite, and cohenite). The origin of phosphide-bearing associations is connected with the effect of highly reduced gases on earlier formed pyrometamorphic rocks.

**Keywords:** mayenite; fluorkyuygenite; Raman spectroscopy; molecular hydrogen; $H_2$; pyrometamorphic rocks; barringerite; shreibersite; Hatrurim Complex; Negev Desert; Israel

## 1. Introduction

Minerals of the mayenite group quite often play a role of rock-forming minerals in larnite and spurrite rocks of the Hatrurim pyrometamorphic Complex [1–4]. The Complex is mainly represented by spurrite, larnite, and gehlenite rocks and also a few types of paralava, spread along the Dead Sea Rift in the territories of Israel, Palestine, and Jordan [5–7]. Commonly, genesis of the pyrometamorphic rocks is connected with combustion processes of organic fuel contained in protolith, but conditions and mechanisms of the formation of the rocks are debatable up to the present [8–14]. Minerals of the mayenite group, fluorkyuygenite, $Ca_{12}Al_{14}O_{32}[F_2(H_2O)_4]$ [1,15], were found in very rare phosphide-bearing pyrometamorphic gehlenite rock in the Negev Desert, Israel. Raman study of fluorkyuygenite indicates the presence of molecular hydrogen ($H_2$) in its structure. In minerals of the fluormayenite-fluorkyuygenite series from pyrometamorphic larnite rocks of the Hatrurim Complex,

$H_2S$ and $HS^-$ groups were specified before [16]. Consequently, $H_2$ is the third neutral molecule after $H_2O$ and $H_2S$, fixed in natural mayenite.

Mayenite, $Ca_{12}Al_{14}O_{32}O$ (*I-43d*, $a \approx 11.95$, $Z = 2$), is an unique microporous material, in which zeolite-like framework $\{Ca_{24}Al_{28}O_{64}\}^{4+}$ is balanced by oxygen $O^{2-}$, occupying statistically two from twelve big (~5 × 7 Å in size) structural cages in the unit cell [17–22]. Free $O^{2-}$ in cages of the mayenite structure can be substituted by $O^-$, $O_2^-$, $O_2^{2-}$, $H^-$, $F^-$, $Cl^-$, $S^{2-}$, $CN^-$, $OH^-$, $Au^-$, etc., and by $e^-$ (electride) too [20,21,23–32]. In synthetic mayenites, hydrogen plays an important role, as it is used in synthesis of electride materials. A pristine $\{Ca_{12}Al_{14}O_{32}\}^{2+}O^{2-}$ is treated by reduced gaseous flow with molecular hydrogen $H_2$ (hydrogenation), which in mayenite dissociates on $H^+$ and $H^-$ [20,21,32–41]. During the UV photoexcitation, a reaction $H^- + \hbar\nu \rightarrow H^0 + e^-$ takes place: in this case, an electron occupies the middle of the cage playing the role of the anion [21,32,33,35,39]. We did not find information about site determination of molecular hydrogen in synthetic mayenite by direct method in literature. Theoretical calculations show that at equilibrium configuration, the $H_2$ molecule occupies the middle position in the cage and the molecule axis is oriented perpendicularly to the Ca-Ca axis ($S^4$ axis), and the H–H distance is 0.764 Å [33]. In an excited state, the $H_2$ molecule turns at an angle to the $S^4$ axis and moves towards the cage wall, and the H-H distance increases up to 0.87–0.92 Å [33]. During polycrystalline electride mayenite heating at 500–750 °C in He atmosphere, $H_2$ escape was observed [21,34]. It is stated that mayenite can be potentially used as a high-temperature sensor for $H_2$ and $O_2$ [41].

In nature, O-dominant mayenite is not known, although a primary mineral named as mayenite from xenoliths in volcanic rocks of Eifel, Germany, was described with formula $Ca_{12}Al_{14}O_{33}$ [42]. Re-investigation of mayenite from Eifel showed that its composition is close to $Ca_{12}Al_{14}O_{32}Cl_2$, and the name of the mineral was changed to chlormayenite [15,43]. Natural minerals isostructural to mayenite are combined to the mayenite supergroup, which includes the two groups: mayenite (oxides) and wadalite (silicates) [15]. The mayenite group is comprised of chlormayenite, $Ca_{12}Al_{14}O_{32}Cl_2$, and fluormayenite, $Ca_{12}Al_{14}O_{32}F_2$, and also their $H_2O$-bearing analogs—chlorkyuygenite, $Ca_{12}Al_{14}O_{32}[Cl_2(H_2O)_4]$, and fluorkyuygenite, $Ca_{12}Al_{14}O_{32}[F_2(H_2O)_4]$ [1,15,40]. Synthetic analogs of $H_2O$-bearing minerals of the mayenite group (kyuygenites) are not known. Kyuygenite forms after water-free mayenite as a result of heated gases containing the water vapor effect [1,4,44]. A hydroxyl group, $OH^-$, is a usual impurity in minerals of the mayenite group; as a rule, it substitutes for fluorine and chlorine: $F/Cl^- \rightarrow (OH)^-$, at central position of the cage. Until the present, the mineral of the mayenite group with formula $Ca_{12}Al_{14}O_{32}(OH)_2$ was not described [1,4,15,44], although its synthetic analog is known [18,22]. A limited amount of the OH groups can enter in mayenite according to the scheme: $O^{2-}$ (O2 site, cage wall) + $(F^-/Cl^-)$ (O3/W site in the cage center) $\rightarrow 3(OH^-)$ (O2a site), accompanied by a partial change of Al coordination from tetrahedral to octahedral [1,20,43].

In the present paper, data on composition and results of Raman study of $H_2$-bearing fluorkyuygenite from Israel are given. This is the first evidence of the molecular hydrogen incorporation to the mayenite structure.

## 2. Materials and Methods

In 2005, at a dry Zohar wadi in the east of the Hatrurim Basin, Negev Desert, Israel M. Murashko found ex situ a big fragment (~0.7 m in diameter) of unusual brecciaed rock, in which barringerite, schreibersite, murashkoite [45], native iron, and shreibersite-iron-cohenite eutectic were detected. In the breccia fragments represented by gehlenite-flamite rock, $H_2$-bearing fluorkyuygenite was revealed. Britvin et al. described the finding as a first barringerite from Israel in 2017 [46] using the samples that Murashko passed to him for investigation. At the end of 2019, we succeeded in finding a bedrock source of the Murashko samples collected more than ten years ago. It is a small channel (5–6 m in width) within low-temperature rocks of the Hatrurim Complex filled with phosphide-bearing breccia. Collected material is studied at present.

Preliminary investigation including mineral identification was performed using thin sections with the help of a scanning electron microscope (Philips XL30/EDAX, Institute of Earth Sciences, Faculty of Natural Sciences, University of Silesia, Sosnowiec, Poland). Chemical analyses of fluorkyuygenite and associated minerals were performed with a CAMECA SX100 microprobe analyzer (Institute of Geochemistry, Mineralogy and Petrology, University of Warsaw, Warszawa, Poland) at 15 kV and 10 nA using the following lines and standards: PK$\alpha$, apatite; ClK$\alpha$, tugtupite; CaK$\alpha$, wollastonite; MgK$\alpha$, SiK$\alpha$, diopside; FeK$\alpha$, hematite; AlK$\alpha$, KK$\alpha$, orthoclase; TiK$\alpha$, rutile; NaK$\alpha$, albite; and FK$\alpha$, fluorphlogopite.

Raman spectra of fluorkyuygenite were recorded on a WITec alpha 300R Confocal Raman Microscope (Institute of Earth Science, Faculty of Natural Sciences, University of Silesia, Sosnowiec, Poland), equipped with an air-cooled solid-state laser (488 nm) and a charge-coupled device (CCD) camera operating at $-61$ °C. The laser radiation was coupled to a microscope through a single-mode optical fiber with a diameter of 3.5 $\mu$m. An air Zeiss LD EC Epiplan-Neofluan DIC-100/0.75NA objective (Carl Zeiss AG, Jena, Germany) was used. Raman scattered light was focused on a broadband single-mode fiber with an effective pinhole size of about 30 $\mu$m, and a monochromator with a 600 mm$^{-1}$ grating was used. The power of the laser at the sample position was changed in the range 20–40 mW. Integration times of 3 s with an accumulation of 20–30 scans were chosen and the resolution was 3 cm$^{-1}$. The monochromator was calibrated using the Raman scattering line of a silicon plate (520.7 cm$^{-1}$). Spectra processing was performed using the SpectraCalc software package GRAMS (Galactic Industries Corporation, Salem, NH, USA). Bands fitting was performed using a Gauss-Lorentz cross-product function, with a minimum number of component bands used for the fitting process.

## 3. Results of Investigation

In brecciaed rock, minerals of the Fe-P-C system are confined to the contact of partially altered flamite-gehlenite rock with fine-grained porous rock which completely altered by low-temperature processes and composed mainly of katoite, calcite, and tacharanite (Figure 1A–C).

The dark parts in the thin section shown in Figure 1A are weakly-altered flamite-gehlenite rock enriched with pyrrhotite, wüstite, and perovskite (Figure 1C), small fragments of which underwent secondary melting (Figure 1D). In these fragments, gehlenite, fluorkyuygenite (secondary after fluormayenite), and fluorkyuygenite (fluormayenite)-flamite eutectic fill in intergranular space between partially melted flamite grains (Figure 1D). Gehlenite has a composition that is close to ideal and it matches to the empirical formula $Ca_{2.01}Al_{1.97}Fe^{2+}_{0.01}Mg_{0.01}Si_{1.01}O_7$. Flamite is represented by high-temperature modification of $\alpha'_H$-$Ca_2SiO_4$, which is stabilized by impurities [47,48]. The studied flamite has the following composition—$(Ca_{1.90}Na_{0.06}K_{0.04})_{\Sigma2}(Si_{0.90}P_{0.10})O_4$. Results of microprobe analyses of fluorkyuygenite are given in Table 1. The empirical formula of fluorkyuygenite, calculated on 26 cations with water specified as the difference of total to 100% and sulphur as $H_2S$, looks as follows: $(Ca_{12.09}Na_{0.03})_{\Sigma12.12}(Al_{13.67}Si_{0.12}Fe^{3+}_{0.07}Ti^{4+}_{0.01})_{\Sigma12.87}O_{31.96}[F_{2.02}Cl_{0.02}(H_2O)_{3.22}(H_2S)_{0.15}\square_{0.59}]_{\Sigma6.00}$.

Raman spectra were obtained on a few tens of grains of fluorkyuygenite. Additional bands at about 850–860 cm$^{-1}$ assigned to flamite inclusions are usually observed in these spectra [48]. Several Raman spectra were received from relatively large (up to 10 $\mu$m) homogeneous grains of fluorkyuygenite. The typical spectrum is shown in Figure 2A,B. For comparison, a spectrum of OH-bearing fluormayenite from larnite rock, Jabel Harmun, Palestine, with simplified empirical formula $Ca_{12}(Al_{13.5}Fe^{3+}_{0.5})_{\Sigma14}O_{32}[F_{1.6}(OH)_{0.4}]_{\Sigma2}$ (S < 0.1 wt.%), is given.

In the Raman spectrum of fluorkyuygenite in the 100-1000 cm$^{-1}$ range, the bands are characteristic for the mayenite framework: ~890 cm$^{-1}$ $\nu_3(AlO_4)^{5-}$, 767 cm$^{-1}$ $\nu_1(AlO_4)^{5-}$, 516 cm$^{-1}$ $\nu_4(AlO_4)^{5-}$ and $\nu(Al$-$O$-$Al)$], 339 cm$^{-1}$ $\nu_2(AlO_4)^{5-}$, and 175 cm$^{-1}$ $\nu(Ca$-$O)$ are present [1,4,15,23,49–52]. The fluormayenite spectrum is similar to the spectrum of the studied fluorkyuygenite. The main differences consist of an appearance of the band at 703 cm$^{-1}$ and a shift of the $\nu_2(AlO_4)$ peak maximum to 317 cm$^{-1}$ (Figure 2). The band at 700–710 cm$^{-1}$ is often present in the spectra of the mayenite group minerals [1,15,44], but it is absent in the spectra of synthetic mayenite [23,49,50,52]. This band

is related to the $\nu_1(Fe^{3+}O_4)^{5-}$ vibrations [53]. The presence of the 2700–3700 cm$^{-1}$ wide band with a few maxima, responding to O-H stretching vibrations in $H_2O$, is a diagnostic property of this mineral, distinguishing it from fluormayenite (Figure 2) [1].

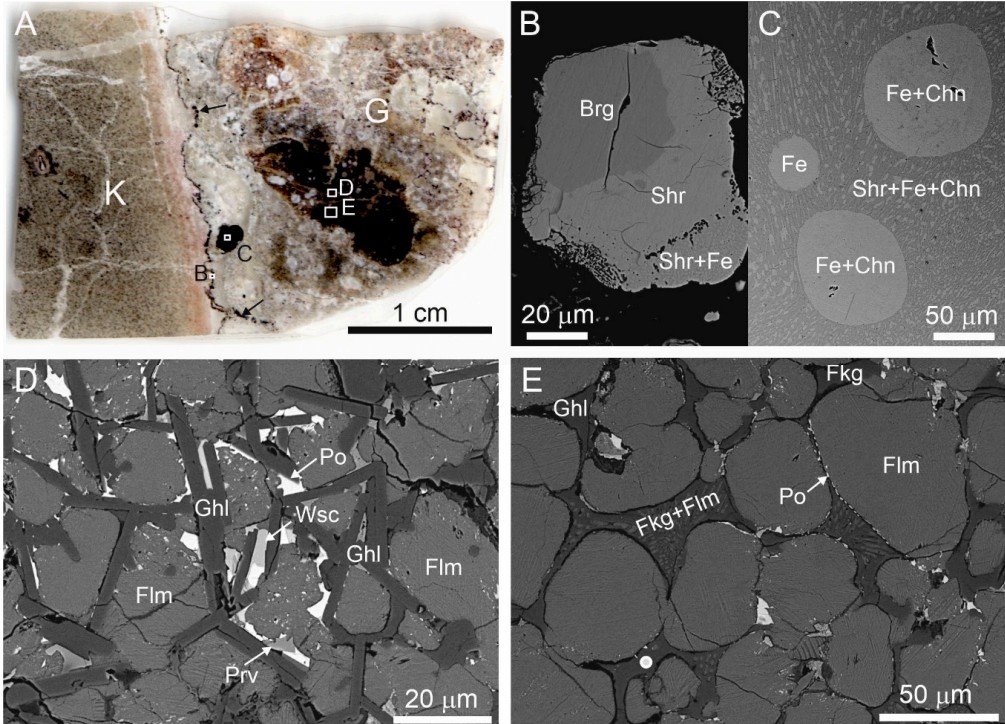

**Figure 1.** (**A**) Scanned image of thin section. Contact of katoite rock (K) and altered gehlenite-flamite rock (G), including weakly-altered fragment (black). A fine black zone enriched with minerals of the Fe-P-C system (black arrows) is on the rocks contact, these minerals form large oval inclusions in gehlenite rock also. Fragments magnified in Figure 1B–D, E are shown by frames. (**B**) Zonal grain with consequent zones: barringerite, schreibersite, and schreibersite-native iron (Fe) eutectic. (**C**) Fe drop and Fe-cohenite eutectic drops in schreibersite-Fe-cohenite eutectic. (**D**) Unaltered gehlenite-flamite rock with abundant pyrrhotite inclusions and rare inclusions of wüstite and perovskite. (**E**) Flamite-gehlenite rock, which is subjected to repeated partial melting, that is reflected in rounded flamite grains and in the crystallization of xenomorphic gehlenite, fluormayenite (replaced by fluorkyuygenite), and fluormayenite (fluorkyuygenite)-flamite eutectic in intergranular space. The white circle is the point where the Raman spectrum shown in Figure 2A,B was obtained. Brg = barringerite, Chn = cohenite, Fe = native iron, Flm = flamite, Fkg = fluorkyuygenite, Ghl = gehlenite, Po = pyrrhotite, Prv = perovskite, Shr = schreibersite, Wsc = wüstite.

In the fluorkyuygenite, spectrum bands at 3574 cm$^{-1}$ (in fluormayenite at 3572 cm$^{-1}$) and 3641 cm$^{-1}$ are related to stretching vibrations in (OH)$^-$ groups, and respectively are connected with the anion site W(O3) at the cage centrum and an additional site O2a on the cage wall [1,4,16,25,51]. The band at 2604 cm$^{-1}$ (in fluormayenite at 2601 cm$^{-1}$, Figure 2) responds to stretching vibrations H-S in the $H_2S$ molecule located at the W site, and before has been noted in the Raman spectra of minerals of the fluormayenite-fluorkyuygenite series from Jabel Harmun, Palestine [16].

For the first time in the Raman spectra of minerals of the mayenite group, a presence of the band at 4038 cm$^{-1}$ was observed (Figure 2A,B), which we interpret as the band related to the stretching vibrations of H-H in the $H_2$ molecule. During the experiment, a shift of this band and a change of its intensity was not noted with a gradual changing of the laser power. Fluorkyuygenite is a colorless mineral in a thin section; under a 488 nm laser beam, any trace of burning was not observed on the surface of the mineral.

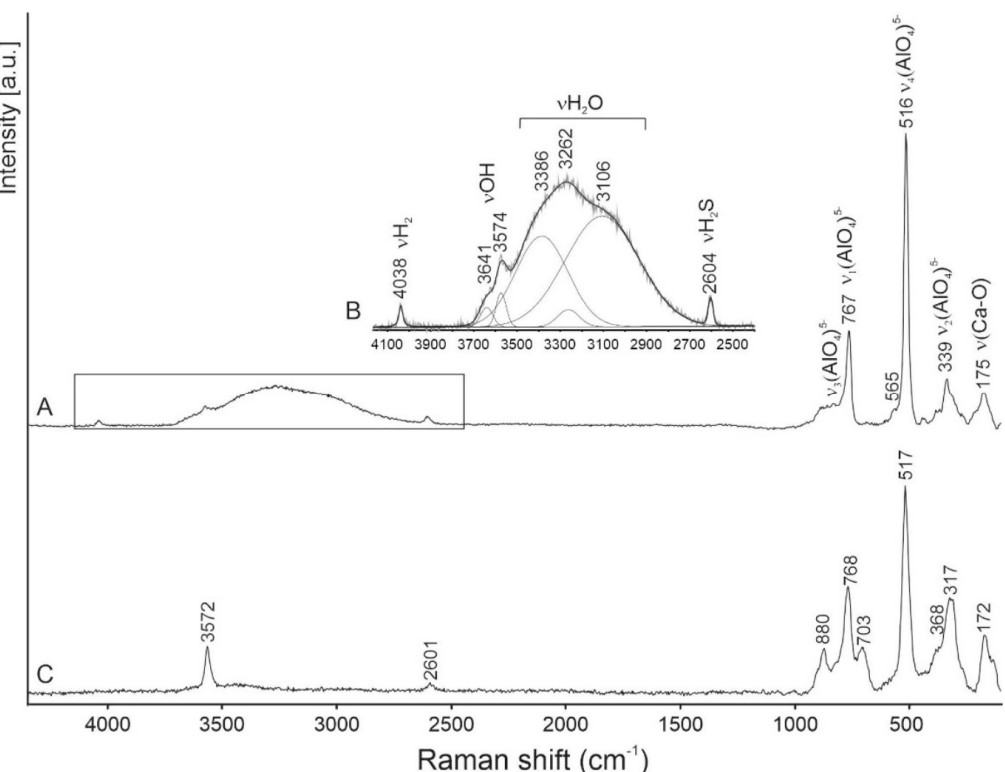

**Figure 2.** (**A**) Raman spectrum of fluorkyuygenite obtained on grain shown in Figure 1E. Fragment magnified in Figure 2B is shown in the frame. (**B**) Raman spectrum of fluorkyuygenite in the 2400–4200 cm$^{-1}$ range. (**C**) Raman spectrum of fluormayenite from larnite rock, Jabel Harmun, Palestine.

**Table 1.** Chemical composition of fluorkyuygenite, wt.%.

|  | *n* = 6 | s.d. | Range |
|---|---|---|---|
| $SiO_2$ | 0.49 | 0.01 | 0.47–0.51 |
| $TiO_2$ | 0.07 | 0.01 | 0.06–0.07 |
| $Fe_2O_3$ | 0.37 | 0.04 | 0.34–0.46 |
| $Al_2O_3$ | 47.31 | 0.13 | 47.11–47.52 |
| CaO | 46.03 | 0.33 | 45.28–46.47 |
| $Na_2O$ | 0.07 | 0.04 | 0.03–0.13 |
| F | 2.61 | 0.12 | 2.45–2.80 |
| Cl | 0.06 | 0.02 | 0.02–0.08 |
| $H_2S$ | 0.34 | 0.05 | 0.25–0.39 |
| $H_2O$ * | 3.93 |  |  |
| -O=F+Cl+S | 1.28 |  |  |
| Total | 100.00 |  |  |
| | Calculated on 26 cations | | |
| Ca | 12.09 |  |  |
| Na | 0.03 |  |  |
| **X** | **12.12** |  |  |
| Al | 13.67 |  |  |
| $Fe^{3+}$ | 0.07 |  |  |
| Si | 0.12 |  |  |
| $Ti^{4+}$ | 0.01 |  |  |
| **T** | **12.87** |  |  |
| F | 2.02 |  |  |
| Cl | 0.02 |  |  |
| $H_2S$ | 0.15 |  |  |
| $H_2O$ | 3.22 |  |  |
| $\square_{vac}$ | 0.59 |  |  |
| **W** | **6.00** |  |  |

* Calculated as difference to 100%.

## 4. Discussion

The role of hydrogen, the most widespread element in our galaxy, is very important in natural processes and so it is considered by scientists investigating global processes of Earth formation and transformation, Earth structure, magmatism, earthquakes, ore deposit genesis, etc. [54–64] and citation therein. Hydrogen in nominally anhydrous minerals from silicate mantle are incorporated as OH-defects, which are determined by a sufficiently high oxygen fugacity, $f$O2, and oxidation of hydrogen to $H_2O$ or OH [65]. Nevertheless, reduced conditions, especially at the early stage of the formation of Earth, could influence $H_2$ incorporation in mantle minerals, which was shown by experimental works [63]. However, in minerals, molecular hydrogen (gas phase) is usually found in fluid and melt inclusions exclusively [66,67]. Gaseous $H_2$ has characteristic bands higher than 4000 $cm^{-1}$: three weak—4126, 4143, and 4161 $cm^{-1}$—and one strong—4156 $cm^{-1}$—in Raman spectra [66,68,69].

FTIR spectroscopy (Fourier-Transform InfraRed Spectroscopy) is mainly used to investigate processes of hydrogen sorption on zeolites and hydrogen dissolution in glass.Taking into consideration that bands of $H_2$ stretching vibrations are practically at the same frequency in Raman and FTIR spectra, results of FTIR spectroscopy can be used for interpretation of Raman spectroscopy data. For example, molecular hydrogen dissolved in vitreous silica displays bands at 4136 $cm^{-1}$ in Raman spectrum and at 4138 $cm^{-1}$ in FTIR spectrum [69]. In Raman spectra of $H_2$-bearing glasses, $Na_2O$-$SiO_2$ and $NaAlO_2$-$SiO_2$, bands assigned to the stretching vibrations of molecular $H_2$ (at ~4125 $cm^{-1}$) and of OH groups (at ~3580 $cm^{-1}$) are present [70]. In experiments on hydrogen sorption at active cation sites in zeolites, bands from stretching vibrations in $H_2$ are at lower frequencies: Ca-zeolite at 4082 $cm^{-1}$ and Mg-zeolite at 4056 $cm^{-1}$ [71–73]. In the structure of natural minerals, $H_2$ was not noted, but in experiments on dissolution of $H_2$ in mantle minerals (clinopyroxene, orthopyroxene, and garnet) at 2–7 GPa and 1100–1300 °C, weak bands from $H_2$ in FTIR spectra near 4062 $cm^{-1}$ were observed. The real position of $H_2$ in the structure of these minerals was not discussed [63].

We did not find, in literature, information concerning Raman investigation of molecular hydrogen in mayenite, both natural and synthetic ones. Raman spectroscopy is successfully used for determination of various O-radicals, OH-groups, $H_2O$, and other groups in mayenite structure, but as a rule, measurements in the interval up to 3700 $cm^{-1}$ are performed [4,49–52]. In the structural cages of fluorkyuygenite, the W site (responds to O3) is mainly occupied by fluorine and molecular water, which set linearly between two Ca (Figure 3A–C) [1]. The calcium sites split and shift to the cage centrum in the case of fluorine at W site, and corresponds to Ca-Ca distance = 4.48 Å (Figure 3B). A presence of bigger water molecules is the reason for the increase of this distance Ca-Ca = 5.28 Å (Figure 3C). In fluormayenite, the Ca-Ca distance = 5.64 Å corresponds to vacancy at the W site [1]. In the studied fluorkyuygenite with the empirical formula $(Ca_{12.09}Na_{0.03})_{\sum12.12}(Al_{13.67}Si_{0.12}Fe^{3+}_{0.07}Ti^{4+}_{0.01})_{\sum12.87}O_{31.96}[F_{2.02}Cl_{0.02}(H_2O)_{3.22}(H_2S)_{0.15}\square_{0.59}]_{\sum6.00}$ $H_2S$ molecule occupies the site analogous $H_2O$ site.

The possible position of the $H_2$ molecule, which responds to theoretical calculations for the equilibrium state of $H_2$ in the mayenite cage [33], is shown in Figure 3D. The distances of Ca-Ca can change in a wide range ~4.48–5.64 Å [1,20] and the position of the $H_2$ molecule is stabilized by a Ca shift towards the middle cage (Figure 3D). That distance correlates well with the Ca-H distance determined for $H_2$ adsorbed at active Ca-sites in zeolites—2.25–3.20Å [72]. To the nearest O at the cage walls of fluorkyuygenite, the H⋯O distance ~ 2.9 Å, which is a few bigger than distances when hydrogen bonds appear [74]. It is supposed that oxygens of the cage can also move towards centrum [1,20], which can influence stabilization of $H_2$ at the cage centrum. There are two equivalent positions of the hydrogen molecule, which are determined by its ideal symmetry and are connected by its rotation around the Ca-Ca ($S^4$) axis. However, it can be assumed that the $H_2$ position in the structural cages is not quite ordered. The band at 4038 $cm^{-1}$, corresponding to stretching vibrations in $H_2$, has a significantly wider full width half maximum (FWHM ~20 $cm^{-1}$, Figure 2B) in the Raman spectrum of fluorkyuygenite in comparison with FWHM (~4 $cm^{-1}$) of the band of gaseous $H_2$ at 4156 $cm^{-1}$ [70], which can point out disordering of $H_2$ in fluorkyuygenite structure.

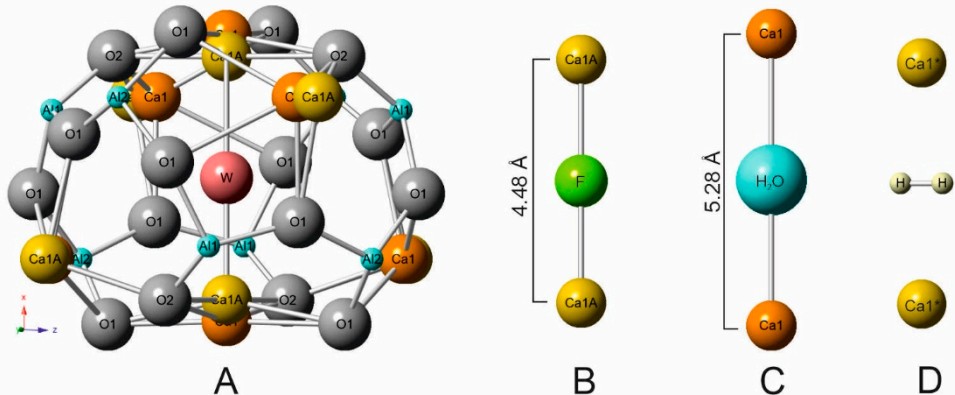

**Figure 3.** (**A**) W site is at the cage centrum in fluorkyuygenite [1]. (**B**) Geometry of linear group Ca-F-Ca. (**C**) Geometry of Ca-H$_2$O-Ca. (**D**) Possible position of H$_2$ molecule stabilized by Ca1* shift towards the cage centrum.

The character of the 4038 cm$^{-1}$ band and its presence in not all Raman spectra of fluorkyuygenite, the repeatability of results, and the absence of visible phase alteration of fluorkyuygenite in the experiment, exclude that this band may be an artefact. It is necessary to underline that in the Raman spectra positions of bands related to HF stretching vibrations set relatively close to the positions of bands from H$_2$. But even in gaseous HF the bands attributed to H-F stretching vibrations are at lower wavenumbers ~3960 cm$^{-1}$ [75]. Consequently, our interpretation of the 4038 cm$^{-1}$ band as the band from the stretching vibration in H$_2$ is more reasonable. The H$_2$ molecule is more linked with the fluorkyuygenite framework compared to H$_2$ adsorbed in zeolites and H$_2$ dissolved in vitreous silica. The band of the H–H stretching vibrations in fluorkyuygenite is at 4038 cm$^{-1}$, which is lower than the position of this band in the Raman spectra of gaseous H$_2$ at ~4156 cm$^{-1}$ [7,9,69] and H$_2$ adsorbed at cation sites in zeolites at ~4050–4100 cm$^{-1}$ [71–73].

The presence of H$_2$ in the structure of natural F-bearing mayenite is connected with reduced and heated gases being by-products of pyrometamorphism at low oxygen fugacity (deep burning), which also defined a formation of phosphides and native iron in brecciaed rocks (Figure 1A–C). Two mechanisms of H$_2$-defects appearing are possible in minerals of the fluormayenite-fluorkyuygenite series: (1) initial capture of H$_2$ from gas and (2) H$_2$ formation of H$_2$O and H$_2$S dissociation.

This is the first report about H$_2$ in natural mayenite. Future systematic Raman spectroscopy investigations involving intervals up to 4200 cm$^{-1}$, and structural investigation of minerals of the mayenite group from pyrometamorphic rocks of different localities, will allow clarification of the mechanism of H$_2$-defect formation.

**Author Contributions:** E.G. and I.G. laboratory investigation, Y.V. and M.M. field trip works. All authors wrote the manuscript. All authors have read and agreed to the published version of the manuscript.

**Funding:** The work was supported by the National Science Centre (NCN) of Poland, grant no. 2016/23/B/ST10/00869.

**Acknowledgments:** 

**Conflicts of Interest:** The authors declare no conflicts of interest.

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
