# Peer review of "Molecular Hydrogen in Natural Mayenite"

_minerals, doi:10.3390/min10060560_

Round 1

Reviewer 1 Report

In the article, the authors have confirmed the existence of molecular hydrogen in natural mayenite via the theoretical calculations as well as Raman spectroscopic results. This is the first time that the presence of molecular hydrogen in synthetic mayenite was confirmed by the direct methods, which is of high importance. The whole paper was well written, and the combined experimental results and theoretical calculations support the conclusion well. For this reason, I recommend the final acceptance of this manuscript.

Author Response

The first reviewer had not remarks.

Reviewer 2 Report

The manuscript describes a possible observation of molecular hydrogen in natural fluorkyuygenite sample using Raman spectroscopy. The observation is very interesting and could have implications on a range of engineering applications of synthetic mayenites. The study is expected to encourage follow-up studies to further investigate the presence of hydrogen molecules in these materials, which may confirm the conclusion here or provide an alternative explanation. Either way, the manuscript is worth to be published in my opinion.

It is worth noting that the frequency assigned to H2 is substantially down shifted comparing with gaseous hydrogen. The shift is larger than that of H2 adsorbed to external or internal material surfaces. This is encouraging since it is consistent with the weakening of H-H bond in a clathrated environment. It should be also noted that the presence of H2 molecule may be related to the speciation of the clathrated anions. In mayenites containing clathrated oxygen ions or electrons, H2 molecules are expected to easily react with them to form OH- or H-, respectively. The material is more likely to incorporate H2 molecules when oxygen in the cages are depleted and replaced with anions like F-/Cl-. It might be worth adding a comment about this.

The manuscript is easy to read in general. I only have one issue. It is unclear how mineralogy is confirmed in this study. There are no diffraction data to support the chemistry-based phase identification. I recommend that the authors clarify this in the revision to give readers more confidence in the conclusion.

Author Response

Responses to reviewer#2:

It is worth noting that the frequency assigned to H2 is substantially down shifted comparing with gaseous hydrogen. The shift is larger than that of H2 adsorbed to external or internal material surfaces. This is encouraging since it is consistent with the weakening of H-H bond in a clathrated environment. It should be also noted that the presence of H2 molecule may be related to the speciation of the clathrated anions. In mayenites containing clathrated oxygen ions or electrons, H2 molecules are expected to easily react with them to form OH- or H-, respectively. The material is more likely to incorporate H2 molecules when oxygen in the cages are depleted and replaced with anions like F-/Cl-. It might be worth adding a comment about this.

It is a correct comment. But clathrated oxygen has not been detected in natural mayenite, we write about it in Introduction. As an answer to this comment we have added term F-bearing” (line 250) to underline a particularity of H2-bearing natural mayenite, in which the framework charge is balanced by F2.       

The manuscript is easy to read in general. I only have one issue. It is unclear how mineralogy is confirmed in this study. There are no diffraction data to support the chemistry-based phase identification. I recommend that the authors clarify this in the revision to give readers more confidence in the conclusion.

Small size of fluorkyuygenite grains does not allow to perform full structural investigation as we did before for different minerals of the mayenite supergroup (Galuskin et al. 2012, 2015; Gfeller et al. 2015). Nevertheless, identification of fluorkyuygenite based on composition and Raman spectra is unmistakable (Galuskin et al. 2016, Środek et al. 2018).     

Reviewer 3 Report

I read the manuscript several times trying to understand the different arguments of the authors to consider their mineral as being enriched by H2 molecules. Finally, I found it suitable for the publication because of several reasons: (i) the Raman data and peak assignments matсh well with previously reported ones as well as the DFT calculations; (ii) mayenite-type compounds are considered as materials for gas storage.

I'm not a specialist in geochemistry and can not be sure that the formation of the minerals and, especially, the oxygen fugacity can lead to the dissociation of either H2O or H2S. However, I found the structural and crystal chemical explanations rather clear and understandable.

Author Response

I'm not a specialist in geochemistry and can not be sure that the formation of the minerals and, especially, the oxygen fugacity can lead to the dissociation of either H2O or H2S. However, I found the structural and crystal chemical explanations rather clear and understandable.

Dissociation of H2O and H2S takes place at heated gases effect. In this case oxygen fugacity is not play any role.  

Reviewer 4 Report

In this work, the authors investigate the composition of H2-bearing fluokyuygenite from Israel as well as the structural features by using Raman spectroscopy. For the first time, the authors assign the 4038 cm-1 Raman band to molecular H2. As the main conclusion, this work evidences the incorporation of molecular H2 in the mayenite structure. The finding is interesting. Minor revision is suggested.

1. The abstract contains very much information that is too specific to be contained in the abstract. I wonder is this a format of this journal. If not, just as a mild suggestion, only the main experiments, and the main conclusion should be included. Those professional details (e.g. specific explanations of the spectrum) can be expanded in the main body.

2. In the Introduction, the authors introduced that the natural mayenite compounds include several isostructural groups and their H2O-bearing analogs. Can the authors provide instances from natural mayenite on its structural stability in the presence of H2O? As indicated by studies on synthetic mayenite, it is not stable in moisture and easily change to Ca3Al2(OH)12 (Inorg. Chem. 2017, 56, 11702−11709). It will be interesting if the natural group shows good H2O resistance.

Author Response

Responses to reviewer#4:

  1. The abstract contains very much information that is too specific to be contained in the abstract. I wonder is this a format of this journal. If not, just as a mild suggestion, only the main experiments, and the main conclusion should be included. Those professional details (e.g. specific explanations of the spectrum) can be expanded in the main body.

We agree with this remark and significantly have reduced Abstract.

  1. In the Introduction, the authors introduced that the natural mayenite compounds include several isostructural groups and their H2O-bearing analogs. Can the authors provide instances from natural mayenite on its structural stability in the presence of H2O? As indicated by studies on synthetic mayenite, it is not stable in moisture and easily change to Ca3Al2(OH)12 (Inorg. Chem. 2017, 56, 11702−11709). It will be interesting if the natural group shows good H2O resistance.

We did not study a stability of fluorkyuygenite and chlorkyuygenite in the presence of H2O. Our experience in natural objects investigation points out that fluormayenite and chlormayenite are more intensively substituted by hydrogarnet than their H2O analogs. A problem of kyuygenite stability in the H2O presence is not within the subject of the present manuscript, so we did not consider that. The studied material after formation did not contact with water.